# Head-to-Head Comparison of [^18^F]FDG PET Imaging and MRI for the Detection of Recurrence or Residual Tumor in Patients with Nasopharyngeal Carcinoma: A Meta-Analysis

**DOI:** 10.3390/cancers16173011

**Published:** 2024-08-29

**Authors:** Natale Quartuccio, Sabina Pulizzi, Domenico Michele Modica, Stefania Nicolosi, Dante D’Oppido, Antonino Maria Moreci, Salvatore Ialuna

**Affiliations:** 1Department of Nuclear Medicine, Ospedali Riuniti Villa Sofia-Cervello, 90146 Palermo, Italy; n.quartuccio@villasofia.it (N.Q.);; 2U.O.C. Otolaryngology, Ospedali Riuniti Villa Sofia-Cervello, 90146 Palermo, Italy

**Keywords:** [^18^F]FDG, nasopharyngeal carcinoma, PET/CT, MRI

## Abstract

**Simple Summary:**

This meta-analysis compared PET imaging (PET/CT and PET) against MRI in detecting a residual tumor or recurrence in patients with nasopharyngeal carcinoma (NPC) at the primary site, retrieving studies that included patients who had both types of scans within a short period after treatment. According to the collected evidence, PET/CT and PET are more sensitive compared to MRI in detecting tumors. However, both methods were equally able to correctly identify patients who did not have tumors. These findings suggest that PET imaging might be more reliable for detecting tumor recurrence or residual disease, which could help doctors make better decisions about patient care and follow-up after treatment for NPC.

**Abstract:**

Background: This meta-analysis compared the diagnostic performance of [^18^F] fluorodeoxyglucose ([^18^F]FDG) positron emission tomography/computed tomography (PET/CT) or PET versus Magnetic resonance imaging (MRI) in detecting recurrence or residual tumors at the primary site in patients with nasopharyngeal carcinoma (NPC). Methods: A comprehensive literature search was conducted in the PubMed/MEDLINE and CENTRAL databases to find studies with at least 20 patients with NPC undergoing both [^18^F]FDG PET/CT (or [^18^F]FDG PET) and MRI for detecting recurrence or assessing residual disease at the primary site. The pooled sensitivity and specificity of PET/CT and MRI were calculated with 95% confidence intervals (CIs) and compared. Results: Five studies, including 1908 patients (six patient groups), were included. PET imaging had higher sensitivity [93.3% (95% CI: 91.3–94.9%); I^2^ = 52.6%] compared to MRI [80.1% (95% CI: 77.2–82.8%); I^2^ = 68.3%], but the specificity of the two modalities was similar: 93.8% (95% CI: 92.2–95.2%; I^2^ = 0%) for PET/CT and 91.8% (95% CI: 90.1–93.4%; I^2^ = 94.3%) for MRI. The areas under the curve (AUCs) for PET/CT and MRI were 0.978 and 0.924, respectively, without significant difference (*p* = 0.23). Conclusions: This meta-analysis suggests that [^18^F]FDG PET imaging and MRI do not significantly differ in diagnostic performance. Nevertheless, [^18^F]FDG PET imaging shows higher sensitivity than MRI.

## 1. Introduction

Nasopharyngeal carcinoma (NPC) is a squamous carcinoma and stands out among head and neck cancers due to its unique anatomical location, epidemiological patterns, and clinical behavior. Approximately 80,000 new cases of nasopharyngeal carcinoma are diagnosed each year, accounting for 0.7% of all cancer cases. In North America and Europe, the incidence rates are below one case per 100,000 individuals. In contrast, endemic regions like Southeast Asia and Southern China (including Hong Kong) have age-standardized incidence rates as high as 20 to 30 cases per 100,000 people annually [1].

Radiotherapy is recommended as the main treatment for primary NPC, and with the development of integrated treatment techniques such as radiotherapy combined with chemotherapy and targeted therapy, the 5-year overall survival (OS) rate for NPC can reach 50–64% [2]. 

NPC is still regarded as an aggressive tumor because of its invasive nature, delayed diagnosis, generally poor prognosis, and low patient survival rate [3]. Indeed, even with recent advancements in treatment procedures and techniques, many patients with nasopharyngeal carcinoma will still face recurrence and ultimately die from locoregional relapse, distant metastases, or both [4]. Approximately 10–20% of patients still experience recurrence at the primary site after the first treatment and improvement of their disease [5]. Recurrent NPC is identified as tumor recurrence following the attainment of complete remission through radical radiotherapy; however, some researchers include both persistent and recurrent disease in their definition of recurrent NPC. “Persistent disease” refers to the presence of a residual tumor at the primary site after initial treatment and generally has a better prognosis than recurrent disease [6].

The interval between the end of treatment and recurrence is variable, but it can amount to only a few months [7]. According to the National Comprehensive Cancer Network (NCCN) guidelines, for patients with resectable head and neck squamous carcinoma who have received radiotherapy, surgery to remove the lesion or local radiotherapy is recommended after recurrence, while chemotherapy alone is usually palliative care for those who are not suitable for radiotherapy or surgery [8]. Re-irradiation therapy may also be a viable alternative for patients whose disease is incurable but whose prognosis characteristics are favorable [9].

Finding optimal treatment strategies for patients with recurrent NPC to prolong their survival after recurrence and to improve their survival and quality of life has been the concern of otolaryngologists and oncologists in recent years [8]. The accurate diagnosis of tumor recurrence or residual disease following treatment is crucial for prompt intervention and improved patient outcomes It is important to detect resectable recurrences early since surgery for these types of tumors can produce excellent outcomes. Therefore, there is a need for reliable imaging techniques to provide surveillance, assess treatment efficacy, and inform subsequent therapeutic decisions. 

Two sophisticated imaging modalities that are frequently utilized in the identification and assessment of cancer are positron emission tomography (PET) and magnetic resonance imaging (MRI). PET imaging gives metabolic and functional information by showing areas of elevated glucose absorption, which is frequently predictive of malignancy [10]. This is especially true when using the radiotracer [^18^F]FDG (fluorodeoxyglucose). On the other hand, MRI is a useful technique for determining the size of tumors and their relationship to adjacent tissues since it provides greater soft tissue contrast and comprehensive anatomical imaging [11]. 

MRI, PET/CT, or PET with [^18^F]FDG are useful modalities in patients with NPC. Making informed treatment options about NPC requires precise identification of any residual or recurring illness. MRI provides enhanced soft tissue contrast, which is vital for delineating tumor boundaries and detecting perineural spread, while [^18^F]FDG PET images offer metabolic insights by highlighting regions of increased glucose metabolism, which is often associated with malignancy, and can offer supplementary information in approximately half of the patients whose MRI results were inconclusive for recurrence [12].

In order to clarify the efficacy of [^18^F]FDG PET imaging and MRI in clinical practice, this manuscript aims to meta-analyze the accuracy of these imaging tests in a head-to-head comparison of the identification of recurrence or a residual tumor in patients with nasopharyngeal cancer.

## 2. Materials and Methods

This meta-analysis was conducted according to the PRISMA guidelines (Preferred Reporting Items for Systematic Reviews and Meta-Analyses) [13]. The study topic, search strategies, inclusion criteria, quality evaluation, data extraction, and statistical analysis were all defined in a protocol that was created prior to the commencement of the literature search. 

### 2.1. Literature Search and Inclusion Criteria

The PubMed/MEDLINE and Central databases were interrogated independently by two researchers (N.Q. and S.P.) to retrieve prospective or retrospective single or multicenter studies carrying out [^18^F]FDG PET/CT and MRI in patients with NPC. The search string used in both PubMed/MEDLINE and Central databases was (PET OR 18F-FDG PET OR positron emission tomography) AND (nasopharyngeal cancer OR nasopharyngeal carcinoma OR cancer of the nasopharynx OR lymphoepithelioma OR NPC) AND (MRI OR magnetic resonance imaging) AND (comparison OR compare OR compared).

No date limit was applied. Articles not in English were excluded. The literature search was updated until 14 May 2024. All identified references were exported to a reference management software (Endnote v. X7.5, Clarivate Analytics).

### 2.2. Study Selection

An investigator reviewed the titles and abstracts of the retrieved entries, selecting only original articles. Duplicates and non-original articles were excluded. The full text of the remaining articles was then reviewed to ensure they met the following inclusion criteria: (1) at least 20 patients with NPC undergoing both [^18^F]FDG PET/CT (or PET) and MRI for detecting recurrence or assessing residual disease at the primary site, (2) a maximum interval of 2 months between PET imaging and MRI, (3) a minimum interval of 2 months between therapy completion and PET imaging, and (4) no history of other malignancies in the patients. Articles with fewer than 20 patients were excluded to reduce the small-study effect, which can result from methodological flaws, outcome reporting bias, and clinical heterogeneity. Additionally, the references of the retrieved articles were screened for further relevant studies.

### 2.3. Data Extraction

From each article, we retrieved the number of true positive (TP), false negative (FN), false positive (FP), and true negative (TN) cases on a per-patient basis for both PET imaging and MRI. For biographical information, the following data were extracted: name of authors, publication year, country, journal, study design, number of patients, clinical setting (detection of recurrence, response assessment, detection of residual tumor), mean age, and sex of patients. 

### 2.4. Methodological Quality Assessment

The methodological quality of the studies was assessed using version 2 of the “Quality Assessment of Diagnostic Accuracy Studies” tool (QUADAS-2), which comprises four domains: patient selection, index test, reference standard, and flow and timing. The concerns about the risk of bias or applicability were described as low, high, or unclear [14].

### 2.5. Statistical Analysis

Statistical analysis was conducted using MetaDisc v. 1.4 (available at http://www.hrc.es/investigacion/metadisc_en.htm, accessed on 14 May 2024) and MedCalc v 19.1.3 (MedCalc Software, Ostend, Belgium; https://www.medcalc.org, accessed on 14 May 2024).

The pooled sensitivity, specificity, Positive Likelihood Ratio (PLR), Negative Likelihood Ratio (NLR), and Diagnostic Odds Ratio (DOR) of PET imaging and MRI, on a per-patient basis, were calculated with 95% confidence intervals (CIs) and compared. A significant difference in performance was determined if the 95% CIs did not overlap between the modalities. The diagnostic accuracy (AC) was calculated by extracting the number of TP, FN, FP, and TN cases from the articles on a per-patient basis. The AC was presented using the summary receiver operating characteristic (SROC) curve and area under the curve (AUC). The AUCs of the two modalities were compared using the method of Hanley and McNeil. 

The I^2^ statistic measured inconsistency, with values of 25%, 50%, and 75% indicating lower thresholds for low, moderate, and high inconsistency due to heterogeneity across the studies. Heterogeneity was assessed at a significance level of *p* = 0.05. The choice between fixed or random effects models was based on the degree of inconsistency, with the random effects model chosen for moderate and high substantial heterogeneity.

## 3. Results

### 3.1. Literature Search

This comprehensive literature search revealed 111 articles (Figure 1). Three entries were excluded as duplicates in the reference manager software. Reviewing titles and abstracts, 102 out of 108 articles were excluded. In particular, among these 102 excluded articles, there were eight case reports, five reviews, and two meta-analyses; the remaining articles were not within the field of interest of the present study. The full texts of the remaining six studies were evaluated to assess the inclusion criteria; one article was excluded because the full text was in Chinese. No additional records were retrieved after crosschecking the references.

Ultimately, the eligibility assessment led to five studies [12,15,16,17,18], including 1908 patients (six patient groups), to be included in the meta-analysis (Table 1 and Table 2). 

### 3.2. Qualitative Analysis of the Studies

All the articles included in the meta-analysis were single-center investigations, published over a large time period (2003–2023), by authors from Europe (n = 1: Italy) and Asia (n = 4: China (3) and Taiwan (1)). Three studies were retrospective, and two studies were prospective. The patient samples were consistently variable across the studies, ranging from 63 to 1463. Three studies used PET/CT and two studies (published in 2003 and 2006) a stand-alone PET scanner. Clinical and imaging follow-up was used as the reference standard in all the studies. The risk of bias for the studies was scored as low by using the QUADAS-2 (Figure 2). Yen et al. first compared the performance of a stand-alone PET scanner and a 1.5-tesla MRI in the detection of recurrence or residual tumor in a retrospective collection of 67 patients after a minimum interval of 4 months from the completion of therapy. Although reporting a perfect sensitivity (100%), the author documented an optimal specificity (93%) for PET imaging. Conversely, MRI documented suboptimal sensitivity (62%) and specificity (43%) [18]. Subsequently, Chan et al. diversified the patient population in a group of patients evaluated for the detection of recurrence and a group evaluated for the assessment of a residual tumor. In both groups, the stand-alone PET and the 1.5-tesla MRI obtained comparable optimal sensitivity and specificity values [12]. Similarly, Ng and co-workers documented equal accuracy for PET/CT (sensitivity: 89%, specificity: 96%) and a 3-tesla MRI (sensitivity: 86%; specificity: 96%) in a prospective group of patients evaluated either for the detection of recurrence or the assessment of a residual tumor [19]. In another study, instead, Comoretto and colleagues reported a trend for better accuracy (92.1%) for MRI compared to PET/CT (85.7%) in a group of 63 patients [15]. The largest comparative study ever published is that of OuYang et al. In a group of 1453 subjects assessed for the detection of recurrence, they documented the superior sensitivity of PET/CT (94%) compared to MRI (79%), although the specificity of the two modalities was similar (94% and 95%, respectively) [17]. 

### 3.3. Meta-Analysis: Diagnostic Performance of PET Imaging and MRI

A random-effects model was chosen for statistical pooling. [^18^F]FDG PET imaging had higher sensitivity [93.3% (95% CI: 91.3–94.9%); I^2^ = 52.6%] compared to MRI [80.1% (95% CI: 77.2–82.8%); I^2^ = 68.3%], but the specificity was similar for both modalities: 93.8% (95% CI: 92.2–95.2%; I^2^ = 0%) for PET imaging and 91.8% (95% CI: 90.1–93.4%; I^2^ = 68.3%) for MRI (Figure 3 and Figure 4). No significant difference was found for PLR, NLR, and DOR between the two imaging modalities. The AUCs for PET/CT and MRI were 0.978 and 0.924, respectively, without a significant difference (*p* = 0.23) (Figure 5). For [^18^F]FDG PET imaging and MRI, the results of the Spearman correlation coefficient demonstrated no threshold effect heterogeneity (Spearman correlation coefficient= 0.42, *p* = 0.39 for PET imaging; Spearman correlation coefficient = −0.39, *p* = 0.78 for MRI).

## 4. Discussion

Nasopharyngeal carcinoma (NPC) has a global incidence of 1 per 100,000 persons per year [20]. Nevertheless, its incidence in occidental countries is lower (especially in the pediatric age) compared to Southeastern Asia (1 vs. 8 per 100,000 persons per year [20,21], and three times higher in men compared to women [21]. The variable incidence of NPC between eastern and occidental countries can be attributed to the different weights of its main risk factors, including salted fish, smoking, and alcohol consumption [22]. Indeed, the majority of clinical studies conducted with NPC patients have been carried out in China and the Republic of Korea [23].

Nasopharyngeal carcinoma may have a clinical presentation with various symptoms, from bilateral nasal obstruction, to bloody rhinorrhea, to ear cotton wool, and up to manifestations with laterocervical metastases. Videoendoscopy is one of the main tests for early detection of the neoformation at the level of the nasopharynx, which is often occupied by ulcerated, vegetating, and irregular neoformation. Endoscopic biopsy is one of the main steps for diagnosis, preceded or not by any radiological diagnostics, mainly MRI. Conventional imaging has often performed suboptimally, particularly during follow-up, because it struggles to distinguish therapy-induced changes from residual disease. Another, yet debated, advantage of PET/CT is its prognostic capability [24]. 

The management of individuals with NPC who have a residual tumor or recurrence at the primary location requires a different approach because of the variations in biological behavior, timing, and available therapeutic choices. Indeed, a residual tumor is one that is discovered right away following initial treatment (such as radiotherapy or chemoradiotherapy) but before the tumor is completely removed [8]. 

A residual tumor has to be evaluated right away after treatment, frequently by imaging (MRI, PET-CT) and biopsy to determine whether the disease is still present. The treatment strategies for a residual tumor include reirradiation, surgery, and chemotherapy. Reirradiation is frequently required when the original radiation was insufficiently effective. To administer large doses without damaging the surrounding normal tissues, methods including stereotactic body radiotherapy (SBRT) and intensity-modulated radiation therapy (IMRT) are used. If the remaining tumor is accessible and re-irradiation is not enough, surgery may be considered. In specific circumstances, an endoscopic nasopharyngectomy may be performed. Chemotherapy may be added if re-irradiation proves insufficient or if there is concern about microscopic illness [6]. 

A recurrent tumor is usually discovered at follow-up appointments, frequently months or years following the first course of therapy. To determine the degree of the recurrence and to confirm it, a restaging workup that includes imaging and potentially a biopsy is conducted. For a recurrent tumor, re-irradiation is an important alternative, although, and because of the cumulative radiation dose from the first treatment, the strategy may be more cautious. Surgery is frequently given more consideration in situations of recurrence, particularly if there was a substantial interval without disease. Depending on the location and size of the tumor, either open surgery or endoscopic surgery may be attempted. Treatment alternatives such as chemotherapy, targeted therapy, or immunotherapy may be given greater consideration. This is particularly the case if the tumor exhibits aggressive behavior or if all other choices have been exhausted [8].

Regarding the significance for prognosis of this discrimination, since the tumor survived the first round of harsh therapy, the term “residual tumor” usually denotes a more resistant malignancy. This frequently calls for more intensive follow-up care and may suggest a more difficult prognosis. Conversely, in the case of a recurrent tumor, the prognosis varies depending on how long it takes for a recurrence and how well the previous therapy worked. In comparison to a residual tumor scenario, the prognosis can be better if the recurrence happens after a protracted period of disease-free living [6,8].

Various studies have compared [^18^F]FDG PET/CT or PET and MRI in detecting residual or recurrent nasopharyngeal carcinoma (NPC). The majority of the studies suggest that [^18^F]FDG PET imaging (irrespective of using PET or PET/CT) may be more accurate than MRI for detecting a residual tumor after therapy or recurrence at the primary site in patients with NPC [12,15,16,17,18]. Among the studies included in this meta-analysis, only Comoretto et al. found a trend toward higher overall accuracy for MRI compared to [^18^F]FDG PET/CT in detecting residual or recurrent NPC at the primary site, with MRI achieving 92.1% accuracy versus 85.7% for [^18^F]FDG PET/CT [15]. Interestingly, also in both the studies using a stand-alone PET, MRI showed an inferior accuracy compared to PET [12,18]. Due to the limited number of the studies, we could not perform a subgroup analysis comparing PET/CT and PET. 

In our meta-analysis, the AUCs for PET/CT and MRI were 0.978 and 0.924, respectively, suggesting excellent diagnostic performance for both modalities. Importantly, the difference in the AUC between PET/CT and MRI was not statistically significant (p=0.23), indicating a similar overall diagnostic accuracy. Our findings suggest also that [^18^F]FDG PET/CT has higher sensitivity compared to MRI for detecting recurrence at the primary site in patients with NPC. Specifically, [^18^F]FDG PET/CT demonstrated a sensitivity of 93.3% (95% CI: 91.3–94.9%), whereas MRI had a slightly lower sensitivity of 80.1% (95% CI: 77.2–82.8%). Sensitivity indicates the ability of a test to correctly identify true positive cases, and the higher sensitivity of PET/CT suggests it may be more effective in this clinical setting. On the other hand, both PET/CT and MRI showed similar specificity, with PET/CT and MRI having specificity values of 93.8% (95% CI: 92.2–95.2%) and 91.8% (95% CI: 90.1–93.4%), respectively. Specificity measures the ability of a test to correctly identify true negative cases, and the comparable specificity between PET/CT and MRI indicates both modalities are similarly reliable in ruling out recurrence or a residual tumor at the primary site. Furthermore, the Spearman correlation coefficient results showed no threshold effect heterogeneity for either PET/CT (Spearman correlation coefficient = 0.42, *p* = 0.39) or MRI (Spearman correlation coefficient = 0.39, *p* = 0.78). This indicates that the differences in sensitivity and specificity between the studies were not to the result of variations in diagnostic thresholds across studies, due to the size or severity of the tumor, strengthening the reliability of our findings.

The main merits of our meta-analysis include a large number of patients (1908) and a head-to-head comparison between PET imaging and MRI analyzing studies including patient groups undergoing both modalities within a short time interval. The evidence in the literature suggests that the ideal time for the first post-treatment PET/CT scan to detect recurrences is between 2 and 4 months after therapy [24]. However, some reports indicate that performing the scan more than 12 weeks after treatment could yield a higher diagnostic accuracy [25]. Therefore, we chose to insert as an inclusion criterion a minimum interval after therapy for [^18^F]FDG PET imaging. 

This meta-analysis also has some limitations, including the small number of studies and heterogeneity among the studies. Despite the inclusion of a substantial dataset comprising 1908 cases from six patient groups in the meta-analysis, the results were heavily influenced by OuYang et al. [17], which contributed 1453 cases (76%) and was the most recent study, published in 2023. The higher sensitivity of PET imaging compared to MRI was primarily driven by the results of OuYang et al., possibly reflecting an acquired large experience of the PET readers in the assessment of patients with NPC. Conversely, the other four studies yielded similar outcomes between PET imaging and MRI. The oldest study in this analysis dates back to 2003 [18], with two of the studies utilizing PET alone rather than PET/CT [12,18]. The evolution of imaging technology over time should be considered, in particular with the passage from a stand-alone PET scanner to the availability of digital PET scanners along with analogical scanners [10], with a trend for a better spatial resolution. The same technological improvement also applies to MRI. Indeed, the performance of MRI in the oldest study by Yen et al. was considerably worse compared to the other studies included in this analysis [18]. In all the articles included in this meta-analysis, the authors were blinded to the results of any prior or subsequent imaging exam. This fact may disagree with a real-world scenario. Furthermore, three out of the six patient groups were composed of mixed patients without discrimination between the assessment of recurrence or a residual tumor. 

We aim in a further study to restrict our analysis, focusing on the diagnostic value and costs of [^18^F]FDG PET/CT imaging alongside MRI in cases of recurrent NPC. This is important because local fibrous tissue hyperplasia and scar formation following radiotherapy for NPC can result in the thickening of the parapharyngeal soft tissue, flattening of the pharyngeal recess and Eustachian opening, and stiffening of the lateral pharyngeal wall, leading to occlusion of the parapharyngeal space and other changes visible on imaging [19]. In such situations, CT or MRI can be limited in their ability to assess treatment efficacy, making it challenging to differentiate between recurrence from non-cancerous tissue. In contrast, PET shows significantly higher uptake of [^18^F]FDG in recurrent or residual nasopharyngeal carcinoma compared to normal tissues and may also lead to saving the patient from the risk of receiving inappropriate treatments [23]. 

In conclusion, [^18^F]FDG PET/CT appears to offer higher sensitivity compared to MRI in this context, although both modalities demonstrate similar specificity and overall diagnostic accuracy, as measured by AUC. These findings support the use of [^18^F]FDG PET/CT and MRI as effective tools in diagnosing the condition, with the choice between them potentially influenced by other factors such as availability, cost, and patient-specific considerations. Since MRI and PET imaging provide synergic information for the clinicians, combining MRI and [^18^F]FDG PET may be even more accurate for detecting local residual/recurrent NPC than using either modality alone. However, the high cost of PET/MRI scanners limits the possibility of obtaining both MRI-derived and PET-derived information with an examination to only a few medical centers.

## 5. Conclusions

This meta-analysis suggests that both imaging modalities are useful for detecting recurrence or residual tumors in patients with a history of NPC. Although the overall diagnostic performance of [^18^F]FDG PET imaging and MRI did not significantly differ, [^18^F]FDG PET/CT or PET showed higher sensitivity than MRI.

## Figures and Tables

**Figure 1 cancers-16-03011-f001:**
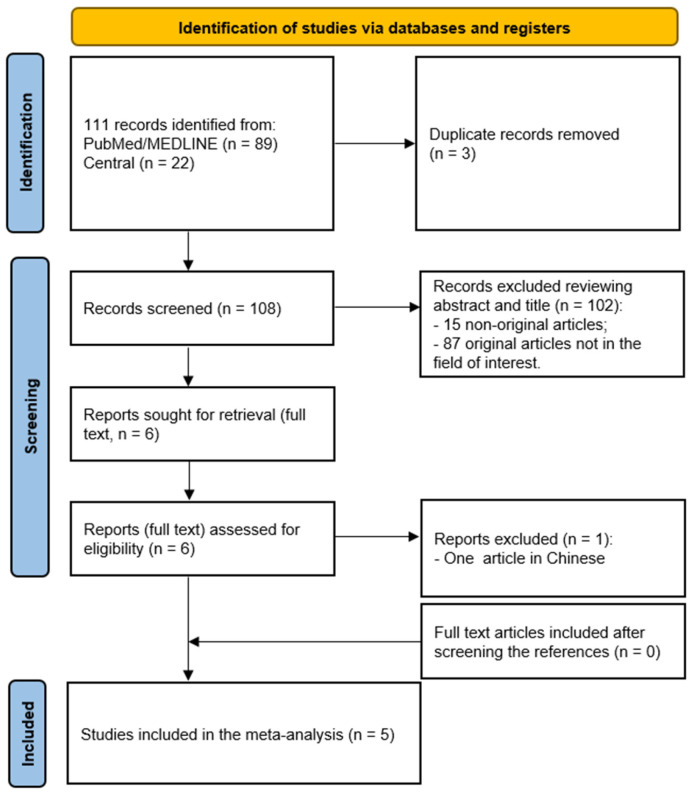
Flow chart of the literature search.

**Figure 2 cancers-16-03011-f002:**
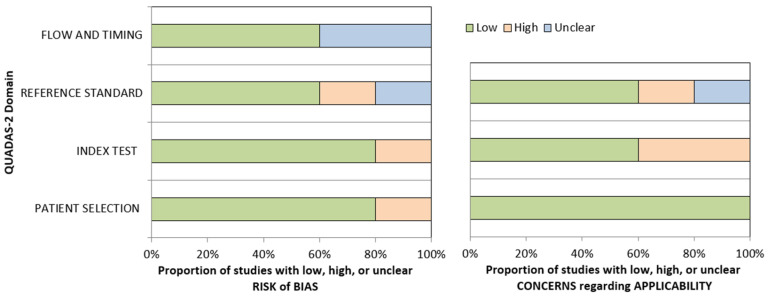
QUADAS 2 assessment results for the included studies of this meta-analysis, indicating low (green color), high (orange color), or unclear (blue color) risk of bias for the relevant domains, and low, high, and unclear concern regarding applicability.

**Figure 3 cancers-16-03011-f003:**
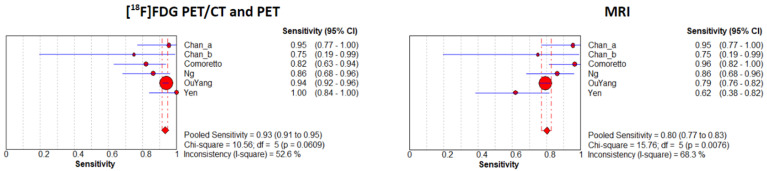
Plots of individual studies and pooled sensitivity of [^18^F]FDG PET/CT or PET and MRI in the detection of recurrence at the primary site or assessment of a residual tumor in patients with NPC on a per-patient-based analysis, including 95% confidence intervals (95% CIs). The size of the squares indicates the weight of each study. Chan_a indicates a patient subgroup evaluated for detection of recurrence; Chan_b indicates a patient subgroup evaluated for the assessment of a residual tumor [12,15,16,17,18].

**Figure 4 cancers-16-03011-f004:**
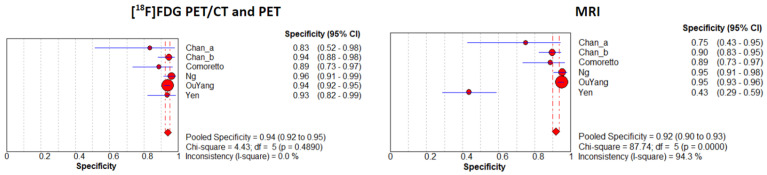
Plots of individual studies and pooled specificity of [^18^F]FDG PET/CT or PET and MRI in the detection of recurrence at the primary site or assessment of a residual tumor in patients with NPC on a per-patient-based analysis, including 95% confidence intervals (95% CIs). The size of the squares indicates the weight of each study. Chan_a indicates a patient subgroup evaluated for detection of recurrence; Chan_b indicates a patient subgroup evaluated for the assessment of a residual tumor [12,15,16,17,18].

**Figure 5 cancers-16-03011-f005:**
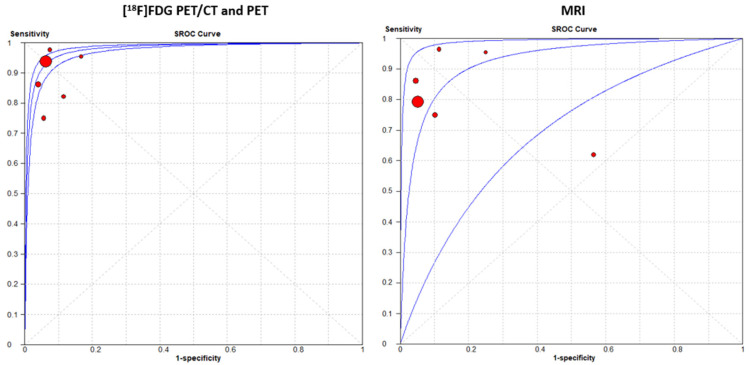
SROC curves of [^18^F]FDG PET/CT or PET and MRI and in the detection of recurrence or assessment of a residual tumor at the primary site in patients with NPC on a per-patient-based analysis.

**Table 1 cancers-16-03011-t001:** Main characteristics of the six patient groups extracted from the 5 studies included in the meta-analysis. P: prospective; R: retrospective. Chan_a indicates a patient subgroup evaluated for detection of recurrence; Chan_b indicates a patient subgroup evaluated for the assessment of residual tumor.

Authors	Year	Country	Journal	Study Design	Number Patients	Interval from Completion of Therapy to Imaging (Months)	Mean Age	Sex (% Male)
Chan_a [12]	2006	China	Eur J Nucl Med	P	34	3	48 ± 11	22/34
Chan_b [12]					112	3	48 ± 12	81/112
Comoretto [15]	2008	Italy	Radiology	R	63	2–14	52	44/63
Ng [16]	2010	Taiwan	Eur Radiol	P	179	3–56	47.9	136/179
OuYang [17]	2023	China	Eur J Nucl Med	R	1453	>6	47	1104/1453
Yen [18]	2003	China	Cancer	R	67	4–70	46.6 ± 12.5	53/67

**Table 2 cancers-16-03011-t002:** Diagnostic performance of PET and MRI in the studies included in this meta-analysis.

Authors	Setting	PET or PET/CT				MRI			
		TP	TN	FP	FN	TP	TN	FP	FN
Chan_a [12]	Detection of recurrence	21	10	2	1	21	9	3	1
Chan_b [12]	Response assessment 3 months after therapy	3	102	6	1	3	97	11	1
Comoretto [15]	Detection of recurrence or residual tumor	23	31	4	5	27	31	4	1
Ng [16]	Detection of recurrence or residual tumor	25	144	6	4	25	143	7	4
OuYang [17]	Detection of recurrence	658	705	47	43	556	713	39	145
Yen [18]	Detection of recurrence or residual tumor	21	43	3	0	13	20	26	8

TP: true positive; TN: true negative; FP: false positive; FN: false negative. Chan_a indicates a patient subgroup evaluated for detection of recurrence; Chan_b indicates a patient subgroup evaluated for the assessment of residual tumor.

## Data Availability

The data supporting the results and conclusions of this article will be made available by the authors on request.

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
