# Peer review of "Head-to-Head Comparison of [18F]FDG PET Imaging and MRI for the Detection of Recurrence or Residual Tumor in Patients with Nasopharyngeal Carcinoma: A Meta-Analysis"

_cancers, 2024, doi:10.3390/cancers16173011_

Round 1

Reviewer 1 Report

Comments and Suggestions for Authors

This study is very interesting. This meta-analysis compared PET/CT against MRI in detecting residual tumor or recurrence in patients with nasopharyngeal carcinoma (NPC) at the primary site. The authors put forward a comprehensive view on how to choose the imaging detection for NPC patients. The comments about this manuscript are as follows.

Comments:

1.Is the study comparing 3 detection methods? PET/CT, PET and MRI?  Please confirm the description in line 10 “This meta-analysis compared PET/CT and PET against MRI in detecting residual tumor or recurrence in patients with nasopharyngeal carcinoma (NPC) at the primary site”

2.I suggest the authors analyze the diagnostic value of [18F]FDG PET imaging and MRI in recurrent nasopharyngeal carcinoma in the further research. Because local fibrous tissue hyperplasia and cicatricial formation after radiotherapy for nasopharyngeal carcinoma may lead to thickening of parapharyngeal soft tissue, flattening of pharyngeal recess and eustachian opening, stiffness of the lateral pharyngeal wall, occlusion of parapharyngeal space and other imaging changes. In this case, CT or MRI has obvious limitations in judging the efficacy, which is difficult to distinguish from the recurrence or survival of nasopharyngeal carcinoma. However, in PET/CT, the uptake of 18F-FDG in recurrent or residual nasopharyngeal carcinoma was significantly higher than that in normal tissues. Combined with the same machine CT images, the recurrent tumor and surrounding tissue fibrosis could be clearly distinguished. 

Author Response

Comment 1: Is the study comparing 3 detection methods? PET/CT, PET and MRI?  Please confirm the description in line 10 “This meta-analysis compared PET/CT and PET against MRI in detecting residual tumor or recurrence in patients with nasopharyngeal carcinoma (NPC) at the primary site”

Response 1: We make this more clear in the lines 10-12 changing the period as follows: "This meta-analysis compared PET imaging (PET/CT and PET) against MRI in detecting residual tumor or recurrence in patients with nasopharyngeal carcinoma (NPC) at the primary site."

Comment 2: I suggest the authors analyze the diagnostic value of [18F]FDG PET imaging and MRI in recurrent nasopharyngeal carcinoma in the further research. Because local fibrous tissue hyperplasia and cicatricial formation after radiotherapy for nasopharyngeal carcinoma may lead to thickening of parapharyngeal soft tissue, flattening of pharyngeal recess and eustachian opening, stiffness of the lateral pharyngeal wall, occlusion of parapharyngeal space and other imaging changes. In this case, CT or MRI has obvious limitations in judging the efficacy, which is difficult to distinguish from the recurrence or survival of nasopharyngeal carcinoma. However, in PET/CT, the uptake of 18F-FDG in recurrent or residual nasopharyngeal carcinoma was significantly higher than that in normal tissues. Combined with the same machine CT images, the recurrent tumor and surrounding tissue fibrosis could be clearly distinguished. 

Response 2: We implemented your comment at the end of the discussion (page 8).  "We aim in a further study to restrict our analysis focusing on the diagnostic value and costs of [18F]FDG PET/CT imaging alongside MRI in cases of recurrent NPC. This is important because local fibrous tissue hyperplasia and scar formation following radiotherapy for NPC can result in thickening of the parapharyngeal soft tissue, flattening of the pharyngeal recess and Eustachian opening, and stiffness of the lateral pharyngeal wall, leading to occlusion of the parapharyngeal space and other changes visible on imaging [J Nucl Med. 2004 Oct;45(10):1669-76]. In such situations, CT or MRI can be limited in their ability to assess treatment efficacy, making it challenging to differentiate between recurrence from non-cancerous tissue. In contrast, PET shows significantly higher uptake of [18F]FDG in recurrent or residual nasopharyngeal carcinoma compared t-o normal tissues, and may lead also to save the patient from the risk of receiving inappropriate treatments. "

Reviewer 2 Report

Comments and Suggestions for Authors

This study performed a meta-analysis to compare the diagnostic performance of residual and recurrent nasopharyngeal carcinoma by using PET/CT and MRI. From an extensive literature search, 6 studies were found eligible, and the included cases could be combined to perform meta-analysis. The results show that PET had higher sensitivity (93.3%) compared to MRI (80.1%), and the specificity was similar, 93.8% for PET/CT and 91.8% for MRI. The methods are described reasonably well, and the results for sensitivity and specificity are clearly presented. Here are a few suggestions:

1.     Although a good dataset of 1,908 cases from six studies was used in the meta-analysis,  it was dominated by OuYang et al. with 1,453 cases (76%), which was also the newest study published in 2023. The oldest study was published in 2003, and two used PET alone, not PET/CT. The change in imaging technology should also be noted, and one may argue that the meta-analysis performed using these 6 identified studies did not add much value. Two observations: (1) The performance in the oldest study, Yen et al. (2003), was much worse compared to others. (2) The higher sensitivity was mainly driven by OuYang et al., and the other 4 studies showed comparable results. I would suggest adding a paragraph to briefly describe these six studies (design, methods and results, and then refer to Table 1 and Table 2), so readers can appreciate the individual results, not dominated by the largest study.

2.      The setting is different – some focused on recurrence, some aimed to assess treatment response, and some included the diagnosis of both residual disease and recurrence. Please give more clear definitions for these 3 types of studies, e.g., when assessing treatment response and progression of residual disease, were the prior imaging studies used as references? These may affect the combined results. When summarizing each of these 6 studies, please pay attention to describing these differences.

3.     The cited results in the fourth paragraph of the Discussion section were not the same as those reported in the Results. Please check carefully and correct them.

4.     Imaging technology is continuously improving over the 2 decades, especially PET/CT, which has much improved spatial resolution. This should be discussed. Given the dominating case number in OuYang et al., and that the final conclusion of this meta-analysis was essentially the same as reported in OuYang et al., what was the value of this meta-analysis? Please discuss and add limitations.

5.     For this study to be more interesting to readers and improve its clinical significance, please add a discussion paragraph to talk about the current management of patients diagnosed with residual disease vs. new recurrence. Given that this meta-analysis did not add much new information, a good clinical discussion will make this article more interesting.

Author Response

Thank you very much for your useful and deep review of our manuscript. Please find the detailed responses below and the corresponding revisions/corrections highlighted/in track changes in the re-submitted manuscript.

Comment1 : Although a good dataset of 1,908 cases from six studies was used in the meta-analysis,  it was dominated by OuYang et al. with 1,453 cases (76%), which was also the newest study published in 2023. The oldest study was published in 2003, and two used PET alone, not PET/CT. The change in imaging technology should also be noted, and one may argue that the meta-analysis performed using these 6 identified studies did not add much value. Two observations: (1) The performance in the oldest study, Yen et al. (2003), was much worse compared to others. (2) The higher sensitivity was mainly driven by OuYang et al., and the other 4 studies showed comparable results. I would suggest adding a paragraph to briefly describe these six studies (design, methods and results, and then refer to Table 1 and Table 2), so readers can appreciate the individual results, not dominated by the largest study.

Reponse 1:  We added your main comment among the limitations of the study, at page 8, in the discussion. "Despite the inclusion of a substantial dataset comprising 1,908 cases from six patient groups in the meta-analysis, the results were heavily influenced by OuYang et al. [18], which contributed 1,453 cases (76%) and was the most recent study, published in 2023. The higher sensitivity of PET imaging compared to MRI was primarily driven by the results of OuYang et al., possibly reflecting an acquired large experience of the PET readers in the assessment of patients with NPC. Conversely, the other four studies yielded similar outcomes between PET imaging and MRI. The oldest study in the analysis dates back to 2003 [15], with two of the studies utilizing PET alone rather than PET/CT [12,15]. The evolution of imaging technology over time should be considered, in particular with the passage from stand-alone PET scanner to and the availability of digital PET scanners along with analogical scanners [10], with a trend for a better spatial resolution. The same technological improvement also applies to MRI. Indeed, the performance of MRI in the oldest study by Yen et al. was considerably worse compared to the other studies included in the analysis [15]" Furthermore, we described in details the design, methods, and results of the studies in a differen paragraph, with references to table 1 and 2 at tpage 5-6.

Comment 2: The setting is different – some focused on recurrence, some aimed to assess treatment response, and some included the diagnosis of both residual disease and recurrence. Please give more clear definitions for these 3 types of studies, e.g., when assessing treatment response and progression of residual disease, were the prior imaging studies used as references? These may affect the combined results. When summarizing each of these 6 studies, please pay attention to describing these differences. D

Reponse 2: Dear reviewer, in all the articles, the author evaluating the images were blinded to the results of any prior or subsequent clinical or imaging study. We added a period in the discussion: In all the articles included in the meta-analysis, the authors were blinded to the results of any prior or subsequent imaging exam. This fact may disagree with a real-world scenario, furthermore, in three out of the 6 patients groups were composed by mixed patients without discrimination between assessment of recurrence or residual tumor.

Comment 3: The cited results in the fourth paragraph of the Discussion section were not the same as those reported in the Results. Please check carefully and correct them.

Response 3: We checked and  corrected the results reported in the discussion.

Comment 4: Imaging technology is continuously improving over the 2 decades, especially PET/CT, which has much improved spatial resolution. This should be discussed. Given the dominating case number in OuYang et al., and that the final conclusion of this meta-analysis was essentially the same as reported in OuYang et al., what was the value of this meta-analysis? Please discuss and add limitations.

Response 4: Dear reviewer the first fact has been highlighted in the discussion in the paragraph "The evolution of imaging technology over time should be considered, in particular with the passage from stand-alone PET scanner to and the availability of digital PET scanners along with analogical scanners [10], with a trend for a better spatial resolution. The same technological improvement also applies to MRI. Indeed, the performance of MRI in the oldest study by Yen et al. was considerably worse compared to the other studies included in the analysis [15]." Furthermore, we added that "The higher sensitivity of PET imaging compared to MRI was primarily driven by the results of OuYang et al., possibly reflecting an acquired large experience of the PET readers in the assessment of patients with NPC. "

Comment 5: For this study to be more interesting to readers and improve its clinical significance, please add a discussion paragraph to talk about the current management of patients diagnosed with residual disease vs. new recurrence. Given that this meta-analysis did not add much new information, a good clinical discussion will make this article more interesting.

Response 5:  Dear reviewer, your obseravation is very good. Indeed, please below an additional part added to the discussion (page 8): "The management of individuals with NPC who have a residual tumor or recurrence at the primary location requires a different approach because of the variations in biological behavior, timing, and available therapeutic choices. Indeed, a residual tumor is one that is discovered right away following initial treatment (such as radiotherapy or chemoradiotherapy) but before the tumor is completely removed [8]. A residual tumorhas to be evaluated right away after treatment, frequently by imaging (MRI, PET-CT) and biopsy to determine whether the disease is still present. The treatment strategies for a residual tumor include reirradiation, surgery and chemotherapy. Reirradiation is frequently required when the original radiation was insufficiently effective. To administer large doses without damaging surrounding normal tissues, methods including stereotactic body radiotherapy (SBRT) and intensity-modulated radiation therapy (IMRT) are used. If the remaining tumor is accessible and re-irradiation is not enough, surgery may be considered. In specific circumstances, endoscopic nasopharyngectomy may be performed. Chemotherapy may be added if re-irradiation proves insufficient or if there is concern about microscopic illness [6]. A recurrent tumor is usually discovered at follow-up appointments, frequently months or years following the first course of therapy. To determine the degree of the recurrence and confirm it, a restaging workup that includes imaging and potentially a biopsy is conducted. For a recurrent tumor, re-irradiation is an important alternative, although, because of the cumulative radiation dose from the first treatment, the strategy may be more cautious. Surgery is frequently given more consideration in situations of recurrence, particularly if there was a substantial interval without disease. Depending on the location and size of the tumor, either open surgery or endoscopic surgery may be attempted. Treatment alternatives such as chemotherapy, targeted therapy, or immunotherapy may be given greater consideration. This is particularly the case if the tumor exhibits aggressive behavior or all other choices have been exhausted [8]. Regarding the significance for prognosis of this discrimination, since the tumor survived the first round of harsh therapy, the term "residual tumor" usually denotes a more resistant malignancy. This frequently calls for more intensive follow-up care and may suggest a more difficult prognosis. Conversely in the case of a recurrent tumor, the prognosis varies depending on how long it takes for a recurrence and how well previous therapy worked. In comparison to a residual tumor scenario, the prognosis can be better if the recurrence happens after a protracted period of disease-free living [6,8]."

Reviewer 3 Report

Comments and Suggestions for Authors

The review systematically summarized the comparison of two advance imaging modalities for NPC.   It is an interesting topic. All sections are concise and well organized and the conclusion is clear.

However, the manuscript needs a minor revision before it is accepted for publication.

Comments:

1.       What is AUC line 29? Make a list of abbreviations at the end of manuscript.

2.       A simple introduction for PET and MR imaging introduction is necessary in the section of introduction in the paragraph “ line 76-83”.

3.       Line 152, ..102/108 articles.. changed to ..102 out of 108 articles..

4.       Table 2,  PET/CT  need to added to PET, as PET or PET/CT even though only a few studies used PET/CT, as showed in Fig3 to Fig 5

5.       Figure 1 ( line 150), Figure 2 (Line 174) and other Fig in the text need to be consistent.

Author Response

Thank you for your useful comments. Please read below our responses.

Comment 1: What is AUC line 29? 

Response 1: AUC stands for area under the curve. We added the full name before the abbreviation. we could not make a list of abbreviations at the end of manuscript because the list is not reccomended in the author guidelines.

Comment 2: A simple introduction for PET and MR imaging introduction is necessary in the section of introduction in the paragraph “ line 76-83”.

Response 2: We added an additional introduction to PET and MRI. "Two sophisticated imaging modalities that are frequently utilized in the identification and assessment of cancer are positron emission tomography (PET) and magnetic resonance imaging (MRI). PET imaging gives metabolic and functional information by showing areas of elevated glucose absorption, which is frequently predictive of malignancy. This is especially true when using the radiotracer [18F]FDG (fluorodeoxyglucose). On the other hand, MRI is a useful technique for determining the size of tumors and their relationship to adjacent tissues since it provides greater soft tissue contrast and comprehensive anatomical imaging."

Comment 3: Line 152, ..102/108 articles.. changed to ..102 out of 108 articles..

Response 3: we changed 102/108 articles in 102 out of 108 articles, as requested.

Comment 4: Table 2,  PET/CT  need to added to PET, as PET or PET/CT even though only a few studies used PET/CT, as showed in Fig3 to Fig 5

Response 4: changed as requested.

Comment 5: Figure 1 ( line 150), Figure 2 (Line 174) and other Fig in the text need to be consistent.

Response 5: changed as requested.